# The Association between Academic Schedule and Physical Activity Behaviors in University Students

**DOI:** 10.3390/ijerph20021572

**Published:** 2023-01-15

**Authors:** Yingyi Wu, Pascal W. M. Van Gerven, Renate H. M. de Groot, Bert O. Eijnde, Jan Seghers, Bjorn Winkens, Hans H. C. M. Savelberg

**Affiliations:** 1School of Nutrition and Translational Research in Metabolism (NUTRIM), Department of Nutrition and Movement Sciences, Maastricht University, 6200 MD Maastricht, The Netherlands; 2School of Health Professions Education (SHE), Maastricht University, 6200 MD Maastricht, The Netherlands; 3Department of Educational Development & Research, School of Health Professions Education, Maastricht University, 6200 MD Maastricht, The Netherlands; 4Faculty of Educational Sciences, Open University of the Netherlands, 6419 AT Heerlen, The Netherlands; 5SMRC Sports Medical Research Center, BIOMED Biomedical Research Institute, Faculty of Medicine & Life Sciences, Hasselt University, 3500 Hasselt, Belgium; 6Division of Sport Science, Faculty of Medicine & Health Sciences, Stellenbosch University, Matieland, Stellenbosch 7602, South Africa; 7Department of Movement Sciences, KU Leuven, 3001 Leuven, Belgium; 8Department of Methodology and Statistics, Care and Public Health Research Institute (CAPHRI), Faculty of Health, Medicine and Life Sciences (FHML), Maastricht University, 6200 MD Maastricht, The Netherlands

**Keywords:** sedentary behavior, physical activity, scheduled education time, self-study time, higher education

## Abstract

Background: University students sit too much, which is detrimental to their physical and mental health. Academic schedules, including scheduled education time and self-study time, may influence their physical activity behaviors. Objectives: To investigate (1) the association between scheduled education time and students’ physical activity levels during weekdays; (2) the association between self-study time and students’ physical activity levels during the weekdays and weekends. Methods: 126 (68 Maastricht University (UM); 58 KU Leuven (KUL)) first-year undergraduate students in biomedical sciences (mean ± *SD* age: 19.3 ± 1.0, BMI: 22.0 ± 3.0, 17% men, 83% women) completed a demographics questionnaire and reported their academic activities with a 7-day logbook. Furthermore, their physical activity behavior was measured with the activPAL monitor for 7 days. Linear mixed models were used to examine the associations between university (UM versus KUL), academic activities (scheduled education time and self-study time), and students’ activity levels. Results: During weekdays, each hour of scheduled education time per day was significantly associated with a 1.3 min decrease of moderate to vigorous physical activity (MVPA) per day. Scheduled education time was not significantly associated with the sedentary time, light-intensity physical activity (LPA), and active sedentary behavior ratio. Each hour of self-study per day was significantly associated with 8 min more of sedentary time per day, 6 min less LPA per day, and 1.3 min less MVPA per day. Self-study time was not significantly associated with active sedentary behavior ratio. During the weekend, each hour of self-study time per day was associated with an additional 17.8 min of sedentary time per day and a reduction of 15.2 min of LPA per day. Self-study time was not significantly associated with the time spent doing MVPA and active sedentary behavior ratio. Conclusions: It could be more effective to change students’ physical activity behaviors during self-study than during scheduled education time. Therefore, offering a study environment that reduces sedentary behavior and promotes light-intensity physical activity, is crucial.

## 1. Introduction

University students show a decrease in physical activity during high school [1,2] and become worse when starting higher education and, thus, tend to adopt a sedentary lifestyle [3]. Sedentary behavior is defined as any waking behavior in a sitting or reclining posture at an energy expenditure ≤ 1.5 METs [4]. Sedentary behavior has been recognized as an independent health risk factor that is associated with type-2 diabetes [5,6], cardiovascular disease [5,6,7], and all-cause mortality [5,6,8]. Thus, it is important to promote physical activity and reduce the time being sedentary.

Because sedentary behavior and physical activity are determined by the setting in which they take place [9], the aforementioned academic setting probably affects the sedentary behavior and physical activity of students. At universities, the students attend lectures, courses, or group discussions, as scheduled. It is the norm and is often mandatory that students sit during class. In an observational study, Chim et al. have shown that each scheduled class hour is associated with nine additional minutes of time spent sedentary [10], which suggests that imposed sitting during classes contributes to students’ sedentary behavior. In addition, university students arrange their self-study time, based on their academic schedules for the rest of the day, and also on the weekends. Self-study time occupies a student’s daily life, and most students tend to do this in a sitting position too, thereby impacting their physical activity behavior as well.

In the current study, Maastricht University (UM) and KU Leuven (KUL) are the chosen research sites, as their educational concept is very different, involving significantly different academic schedules, and thus contact hours, for their students. UM has adopted problem-based learning as its educational format, which typically includes fewer scheduled contact hours than the lecture-based curriculum applied at KU Leuven. Consequently, KU Leuven has about twice as many scheduled contact hours as UM. Based on previous research [10], we hypothesize that scheduled education time is related to lower activity levels and a more sedentary behavior. Similarly, we hypothesize that self-study time has a negative relationship with physical activity and a positive relationship with sedentary behavior.

## 2. Methods

### 2.1. Participants

First-year undergraduate, biomedical sciences students were recruited at UM and KUL. Recruitment took place by means of posters, flyers, and 3-min pitches during tutorial group meeting and lectures. Participants having musculoskeletal discomfort or other pathologies that would influence daily physical activity were excluded. Participants were voluntary to participate in this study. They had the right to withdraw at any time without an explanation, and their personal data was kept private and confidential.

### 2.2. Materials

Participants completed a demographics questionnaire about age, gender, height, weight, commuting to university, and gym/sports membership status. To record their scheduled education time and self-reported study time per day, participants completed a structured logbook each day for 7 consecutive days. This logbook offered a daily table with all of the categories, including courses on campus, courses via livestream, lab lessons on campus, lab livestream sessions, self-study time at their residence, and self-study time on campus (library, study room, etc.). They were requested to write down the specific time they had spent on the activities they engaged in each day.

To measure participants’ free-living, physical activity behavior, they wore an activPAL activity sensor for 7 days continuously. The activPAL has been shown to yield reliable and valid measurements of physical activity and sedentary behaviors for the adult population [11]. The activPAL data were processed by the PALbatch software package (version 8.10.9.43, PAL Technologies 28d, Glasgow, UK). The CREA algorithm (enhanced analysis algorithm: non-wear, upright correction, lying, cycling, seated transport) was used when processing the data [12]. ActivPAL data were included in the data analysis if three or more days were classified as valid by the CREA algorithm.

### 2.3. Design and Procedure

This study utilized a cross-sectional, observational design. Measurements took place during two periods: from November to December 2020, and from January to March 2021. During both periods, a comparable number of participants were measured at both universities. Due to the COVID-19 situation, UM and KUL both adopted hybrid education, combining online and onsite teaching in both periods. The rules regarding the lock-down were comparable for the Netherlands and Belgium, and thus for UM and KUL.

Study information brochures and informed consent forms were given to each participant, and the researcher also verbally informed potential participants and answered all of the questions they had before the instruction day. Once signed informed consent was received, the study started. On the instruction day, the researcher prepared all of the materials and the sensors in advance and met with participants in a meeting room. Upon completing the demographic questionnaire, the activPAL was activated and attached to the middle-anterior of the participants’ right thigh, which registered their physical activity behavior for 7 consecutive days. The researcher gave each participant a 7-day logbook, which the participants had to fill in themselves for the following week. The meeting took approximately 15 min per participant.

### 2.4. Data Processing

One hundred and twenty-nine participants were originally included in this study. ActivPAL data of one UM participant was excluded because of technical errors. Two UM participants provided no logbook information and three participants provided no demographic information. Thus, 126 participants, with 68 at UM and 58 at KUL, completed all of the measurements. If participants did not wear the activPAL for four continuous hours or more, the CREA algorithm classified the corresponding day as a non-valid day for that student. Based on this rule, 65 of 68 participants from UM had seven valid days of activPAL data; 52 of 58 participants from KUL had 7 valid days of activPAL data. Three participants from UM had 6 days of activPAL data; four participants from KUL had 6 days of activPAL data, and one UM participant had 4 days and one KUL participant had 3 days of activPAL data.

### 2.5. Independent and Dependent Variables

The independent variables were university (UM, KUL), scheduled education time, and self-study time. Scheduled education time is defined as the sum of the time spent in courses and labs on campus, and the time spent in courses and labs via livestream for each day. Self-study time is the sum of the time spent on self-study at home and other places (e.g., library, study room) for each day.

The dependent variables were sleeping time, sedentary time, active sedentary behavior ratio, LPA, and MVPA. Sedentary time is a combination of sitting time and secondary lying time (e.g., relaxing on a sofa) during each day. Sleeping time is primary lying time in the CREA algorithm. Active sedentary behavior ratio = total duration of sitting bouts lasting ≤ 30 min/total duration of sedentary time. The active sedentary behavior ratio is considered as an indicator of actively interrupted sitting behavior. The higher the ratio, the more time spent sedentary in bouts of 30 min, maximum. The 30-min cut-off is based on the recommendation to interrupt sedentary behavior every 30 min [13]. Moderate-to-vigorous physical activity (MVPA) duration was taken from stepping time with a cadence of ≥100 steps/min [14]. Light intensity physical activity (LPA) was below the threshold of MVPA and above the threshold of sedentary behavior. Therefore, the LPA duration consisted of standing and stepping time with a cadence of <100 steps/minute. Cycling and seated transportation were reported as an output variable by the CREA algorithm. Cycling is a part of the total physical activity, while seated transportation is a subsection of sedentary time.

### 2.6. Statistical Analyses

All statistical analyses were performed using IBM SPSS Statistics for Windows (version 27.0, Armonk, NY, USA). *p* values ≤ 0.05 (two-sided) were considered statistically significant. We calculated that 64 participants were needed per university to detect a medium effect size (*d* = 0.5) in sitting time with 80% power and a significance level α of 5% [15]. Accounting for a drop-out rate of 10%, our initial target was 72 participants per group. Due to the COVID-19 pandemic, the number of participants from KUL was slightly lower than the target sample size.

Independent-sample *t* tests were used for the comparisons of the numerical variables (age, body-mass index (BMI), scheduled education time, self-study time, and physical activity behavior variables) between UM and KUL. Fisher–Freeman–Halton test were used to check the difference in the categorical variables (perceived physical health and perceived physical fitness) between the two universities. Marginal models for repeated measures were used, where different covariance structures of the repeated measures were considered and the one with the smallest Bayesian information criterion (BIC) was chosen. For weekday data, the models examined the associations of university (UM versus KUL), scheduled education time (SET), and self-study time (SST) with each dependent variable, correcting for each day (Monday–Friday). For weekend data, the same models were used, except for the SET, as there is no education scheduled on those days. The results of the final models were reported, i.e., estimated coefficients of the explanatory variables with their corresponding 95% confidence intervals (CI) and *p* values. The linearity assumption for the numerical explanatory variables was checked using scatterplots before analyzing the marginal models.

## 3. Results

### 3.1. Participant Characteristics

The demographic characteristics of the participants are presented in Table 1. UM students were statistically slightly older and had a higher BMI than KUL students. Both universities had a higher proportion of females. KUL students rated themselves higher for perceived physical health (*p* < 0.001) and perceived physical fitness (*p* = 0.017) than UM students.

### 3.2. Self-Reported Scheduled Education Time and Self-Study Time

KUL students reported, on average, 1 h and 45 min more scheduled education time per weekday than UM students (Table 2). UM students reported more self-study time during weekdays and weekends, compared to KUL students. No significant difference in the total time spent on school work was found.

### 3.3. Average Physical Activity Behavior

The mean values (*SD*) of the active sedentary behavior ratio, sedentary time, LPA, MVPA, sleeping time, cycling, and seated transportation were reported separately on weekdays and weekends for UM and KUL students, as displayed in Table 3. The estimated mean differences (model coefficients) between UM and KUL, accounting for each day by using a marginal model for repeated measures, are also presented for the active sedentary behavior ratio, sedentary time, LPA, and MVPA in Table 3.

Sleeping time during the weekdays and weekends was comparable for UM and KUL students. UM students spent significantly less time on seated transportation than KUL students on weekdays and weekends, respectively. Following a correction for the day (marginal model for repeated measures), there was a significant university effect doing MVPA (*p* = 0.045), showing UM students spent 6.4 min/day less than KUL students doing MVPA during weekdays. On weekdays, no significant differences were found for spending time being sedentary, doing LPA, or active sedentary behavior ratio between UM students and KUL students. On weekends, there is a significant university effect on sedentary time (*p* = 0.005), active sedentary behavior ratio (*p* < 0.001), and doing MVPA (*p* = 0.039). During the weekend, UM students spent, on average, 55.8 min more in sedentary time per day, 8.6 min less doing MVPA per day, and had a 0.14 lower active sedentary behavior ratio per day than KUL students.

### 3.4. Association between Scheduled Education Time/Self-Study Time and Students’ Physical Activity Levels

To check the association between scheduled education time/self-study time and students’ physical activity levels, marginal models for repeated measures were used on weekdays and weekends, separately. Considering participants at both universities were first-year students in a comparable age group, and who were not overweight or obesity, age and BMI were not included in the models. Though we found that KUL students reported a higher self-rated physical health and physical fitness than UM students, physical health and physical fitness were not entered into the model because they were not objectively measured.

During weekdays, scheduled education time was not statistically significantly associated with sedentary time, LPA, and the active sedentary behavior ratio. There was a significant correlation between scheduled education time and MVPA (*p* = 0.044; Table 4). That is, each hour of scheduled education time was associated with a 1 min and 18 s decrease in MVPA per day. There were significant correlations between self-study time and sedentary time, LPA, and MVPA (*p = 0*.006, *p* = 0.001, and *p* = 0.019, respectively). Each hour of self-study time per day was associated with an additional 8 min of sedentary time, a decrease of 6 min in LPA, and a decrease of 1 min and 18 s in MVPA per day. The active sedentary behavior ratio was not significantly associated with self-study time. During the weekend, time spent doing MVPA, and active sedentary behavior ratio were not significantly associated with self-study time. Self-study time was significantly associated with sedentary time (*p* < 0.001) and LPA (*p* < 0.001). Each hour of self-study time was associated with an additional 17 min and 40 s in sedentary time per day and a reduction of 15 min and 12 s doing LPA per day.

## 4. Discussion

This study investigated the relationship between academic activities and biomedical students’ physical activity levels at two universities. On weekdays, first-year students of the two universities spent 9 h and 40 min per day being sedentary. This result was consistent with the finding from a systematic review [16] showing that university students, on average, spend 9.8 h sitting per day. We expected that university students’ sedentary behavior is partly caused by the educational setting because students are required to sit when attending lectures and tutorials and spontaneously sit when studying. Therefore, we hypothesized that scheduled education and self-study time is positively associated with sedentary time and negatively associated with physical activity.

We tested this hypothesis using data from two universities (UM and KUL) with different educational systems. These data confirmed that biomedical students at UM have less scheduled education time than the students at KUL, even during the COVID-19 pandemic. UM students reported spending an average of 1 h and 7 min per day, while KUL students reported spending an average of 2 h and 52 min per day on scheduled education time. From their logbook, UM students reported about one hour more of self-study time than KUL students reported, on weekdays and weekends, respectively. Due to the problem-based learning model [17], UM students were expected to use more self-regulated learning and contextual materials searching, based on different problem cases, which may also lead to an increase in self-study time on the weekdays and also during the weekends. Interestingly, the total school work hours were comparable between UM and KUL, which suggests that the study programs at the two universities have a similar study load. Regarding the allocation of study time, KUL students spent most of their time on academic activities (SET and SST) on the weekdays. However, UM students seem to have the opportunity to distribute these academic activities over the entire week. In other words, compared to KUL students, the study load is partially redistributed to the weekend for UM students. As a consequence, UM and KUL students did not differ in sedentary time and LPA during the weekdays, while UM students showed more sedentary time and less actively interrupted sitting on the weekends, due to the redistribution of SST.

We noticed that there was no scheduled education at both universities on the weekend. What’s more, students spent more time on self-study than on scheduled education, both on weekdays and on the weekend. These showed that self-study time made up a larger proportion than scheduled education time, in general, which could partly explain why self-study time had a statistically significant correlation with increasing sedentary behavior and decreasing LPA on the weekdays and the weekends.

It is important to note that this study was conducted during the COVID-19 pandemic. This means that scheduled education time was hybrid, with both online and offline classes, which was different from the situation before the pandemic, when classes were exclusively offline. The way students commute to and from the university could be active (e.g., cycling or walking) or inactive (i.e., seated transportation), which was influenced by onsite scheduled education. Though the two universities showed comparable COVID regulations and a similar hybrid education approach, it was difficult to accurately determine how the commuting behavior of students was different from the normal situation. For this reason, we may have underestimated the role of scheduled education time on the physical activity behavior. Although this is a potential limitation of the current study, it also gives insights into the physical activity and sedentary behavior among university students during the pandemic.

In this study, we have shown that, irrespective of the university’s didactic format, students spend a similar amount of time studying and choose to be seated no matter what type of educational activity, scheduled education, or self-study, they engage in. Sedentary behavior is considered as a health risk factor that is independent of MVPA [18,19]. Therefore, it is essential to reduce sedentary time, irrespective of the time students spend on MVPA. A strategy recommended by the World Health Organization is to replace prolonged sitting with LPA [20]. To achieve that, universities should focus on reducing sitting time during both scheduled education and self-study. According to the current results, it is more effective to change students’ physical activity behavior during self-study time.

## 5. Conclusions

It is important to stimulate students to find ways of balancing academic activities with a more active and less sedentary lifestyle. This means that we should focus more on self-study time than on scheduled education time, and focus more on LPA than on MVPA. Therefore, it is warranted that academic institutions offer university students a study environment (i.e., library or study room) and self-management strategies that not only facilitate academic performance, but also support the students to reduce their sedentary behavior and promote LPA.

## Figures and Tables

**Table 1 ijerph-20-01572-t001:** Demographic characteristics of UM and KUL students.

	Overall (*n* = 126)	UM (*n* = 68)	KU Leuven (*n* = 58)	*p* Values
Age (years) ^†^	19.3 ± 1.0	19.6 ± 1.1	18.9 ± 0.8	<0.001
BMI (kg/m^2^) ^†^	22.0 ± 3.0	23.1 ± 3.2	20.7 ± 2.1	<0.001
Gender (*n*)				
Men	21 (17%)	10 (15%)	11 (19%)	
Women	105 (83%)	58 (85%)	47 (81%)	
Self-rated physical health (*n*) ^‡^				<0.001
Very poor	0	0	0	
Poor	3 (2%)	3 (4%)	0	
Average	34 (27%)	28 (41%)	6 (10%)	
Good	79 (63%)	35 (52%)	44 (76%)	
Very good	10 (8%)	2 (3%)	8 (14%)	
Self-rated physical fitness (*n*) ^‡^				0.017
Very poor	2 (2%)	2 (3%)	0	
Poor	10 (8%)	9 (13%)	1 (2%)	
Average	46 (36%)	28 (41%)	18 (31%)	
Good	58 (46%)	25 (37%)	33 (57%)	
Very good	10 (8%)	4 (6%)	6 (10%)	
Gym membership (*n*)	71 (56%)	41 (60%)	30 (52%)	

Note. *M* ± *SD. M* = mean. *SD* = standard deviation. *n* = sample size. ^†^ independent sample *t* test. ^‡^ Fisher–Freeman–Halton test.

**Table 2 ijerph-20-01572-t002:** Average scheduled education time during the weekdays and average self-study time on weekdays and weekends at UM and KUL.

	UM (*n* = 68)	KU Leuven (*n* = 58)	*p* Value
Average scheduled education time per weekday (hour: minutes)	1:07 (0:45)	2:52 (0:55)	< 0.001
Average self-study time (hour: minutes)			
per weekday	3:22 (1:25)	2:37 (1:15)	0.002
per weekend	3:43 (2:13)	2:26 (2:10)	0.001
Total time spent on school work per week	29:40 (10:05)	32:42 (10:29)	0.103

Note. *M* (*SD*). *M* = mean. *SD* = standard deviation. *n* = sample size. Independent-sample *t* test.

**Table 3 ijerph-20-01572-t003:** Students’ average physical activity behavior data at UM and KUL.

PAB Variables	UM (*n* = 68)	KU (*n* = 58)	*p* Value ^†^	Model Coefficients (KU Leuven as Reference)	95% CI	*p*-Value ^‡^
Active sedentary behavior ratio						
per weekday	0.37 (0.13)	0.37 (0.10)	0.750	−0.02	−0.06, 0.03	0.434
per weekend	0.37 (0.16)	0.51 (0.17)	<0.001	−0.14	−0.20, −0.08	<0.001
Sedentary time (hours:minutes)						
per weekday	9:39 (1:31)	9:41 (1:18)	0.879	0.8 min	−30.9, 32.5	0.960
per weekend	8:54 (2:01)	7:41 (1:50)	<0.001	55.8 min	16.7, 94.8	0.005
LPA (hours:minutes)						
per weekday	3:57 (1:14)	3:36 (0:48)	0.059	20.2 min	−3.3, 43.7	0.091
per weekend	4:07 (1:48)	4:21 (1:24)	0.425	3.4 min	−30.2, 37.0	0.842
MVPA (hours:minutes)						
per weekday	0:18 (0:15)	0:23 (0:19)	0.111	−6.4 min	−12.6, −0.1	0.045
per weekend	0:18 (0:21)	0:28 (0:26)	0.014	−8.6 min	−16.7, −0.5	0.039
Sleeping time (hours:minutes)						
per weekday	9:37 (1:06)	9:28 (0:51)	0.360			
per weekend	10:12 (1:36)	10:25(1:11)	0.384			
Cycling (hours:minutes)						
per weekday	0:08 (0:09)	0:12 (0:14)	0.077			
per weekend	0:06 (0:10)	0:09 (0:16)	0.170			
Seated transportation (hours:minutes)						
per weekday	0:20 (0:32)	0:38 (0:40)	0.005			
per weekend	0:21 (0:31)	0:55 (0:41)	< 0.001			

Note. *M* (*SD*). *M* = mean. *SD* = standard deviation. *n* = sample size. PAB = physical activity behavior. LPA = light-intensity physical activity. MVPA = moderate-to-vigorous physical activity. 95% CI *=* 95% confidence interval. min = minutes. ^†^ Independent-sample *t* test (uncorrected for each day, averaged for weekdays and weekends); ^‡^ marginal model for repeated measures (corrected for each specific day and accounting for the correlation between repeated measures).

**Table 4 ijerph-20-01572-t004:** Association between scheduled education time/self-study time and students’ physical activity levels.

Day Type	Outcomes	Model Coefficients	95% CI	*p* Value
weekdays	Sedentary time (min)			
	SET (hour)	5.9	−0.7, 12.7	0.081
	SST (hour)	8.0	2.3, 13.7	0.006
	Active sedentary behavior ratio			
	SET (hour)	−0.005	−0.012, 0.003	0.227
	SST (hour)	−0.001	−0.007, 0.005	0.756
	LPA (min)			
	SET (hour)	−3.8	−8.1, 0.4	0.077
	SST (hour)	−6.0	−9.8, −2.4	0.001
	MVPA (min)			
	SET (hour)	−1.3	−2.5, −0.1	0.044
	SST (hour)	−1.3	−2.3, −0.2	0.019
Weekends only for SST (hour)	Sedentary time (min)	17.8	11.0, 24.5	<0.001
	Active sedentary behavior ratio	−0.003	−0.012, 0.006	0.523
	LPA (min)	−15.2	−21.1, −9.3	<0.001
	MVPA (min)	−0.2	−1.8, 1.4	0.804

Note. LPA = light intensity physical activity. MVPA = moderate-to-vigorous physical activity. SET = scheduled education time. SST = self-study time. 95% CI = 95% confidence interval. min = minutes.

## Data Availability

The data that support the findings of this study are available from the corresponding author upon reasonable request.

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
