# Peer review of "The Association between Academic Schedule and Physical Activity Behaviors in University Students"

_ijerph, 2023, doi:10.3390/ijerph20021572_

Round 1

Reviewer 1 Report

The article contains a quantitative analysis of the results of research on the relationships between students' physical activity and the schedule of classes at the university and their own learning. Compared to the quantitative characteristics of the research results, the conclusions are very short and general. The conclusions should indicate what universities can offer students to develop physical activity of students during their stay at the university, to take care of their health. The cited publication by the authors is missing from the bibliography: Owen, et. al., 2011.

Reviewer 2 Report

Paper: The Association between Academic Schedule and Physical Activity Behavior in University Students

Dear Authors, your paper is well written. Your results should be published to the societal benefit of battling the problems of physical inactivity of students. There are in my view three flaws in the paper:

1.       The protocol is not clear to me. Below you find details about my confusion in remarks  3 – 5;

2.       The literature seems incomplete. I guess that these are relevant: [1–3], and there are more relevant papers in the IJERPH itself, I think;

3.       Your results should be displayed graphically. I recommend a polar plot, or radar chart, with results along the rays. The picture I want cannot be copied here, from https://medium.com/@DamianC_/football-data-visualisation-womens-world-cup-2019-f025c1009805 . But instead of pressures, ball recoveries, shots, etc. you have other parameters. 

Such plots can be made with MathLab, or Maple or even with Excel. For explanation of the idea see: http://seeingdata.org/taketime/inside-the-chart-radar-chart/

Remarks:

1.       Everywhere: references in the IJERPH journal should be numbered [1], [2], please convert the references to IJERPH format.

2.       In lines 88-89 you explain “participants completed a structured 7-day logbook”. Does this mean that only once during a week was measured?

3.       Line 95 you say that an activity sensor was worn for 7 days continuously, was this once for a week? Or were there two periods of a week?

4.       Lines 112, 113, what does “measurement day” mean here? Measurements are continued and are not restricted to one single day.

5.       Line 118, is unclear. You say “participants had to fill in themselves for the following week.” Your experiment – however - lasted for months. What happens in the weeks thereafter with regard to the week schedule?

6.       Over seven of your references have incomplete author name list, for instance.[4], [5], etc.

References

1.        Duan, J.; Hu, H.; Wang, G.; Arao, T. Study on Current Levels of Physical Activity and Sedentary Behavior among Middle School Students in Beijing, China. PLoS One 2015, 10, e0133544.

2.        Hallal, P.C.; Andersen, L.B.; Bull, F.C.; Guthold, R.; Haskell, W.; Ekelund, U.; Alkandari, J.R.; Bauman, A.E.; Blair, S.N.; Brownson, R.C.; et al. Global physical activity levels: Surveillance progress, pitfalls, and prospects. Lancet 2012, 380, 247–257.

3.        Han, H.; Gabriel, K.P.; Kohl, H.W. Application of the transtheoretical model to sedentary behaviors and its association with physical activity status. PLoS One 2017, 12, e0176330.

4.        Ekelund, U.; Tarp, J.; Steene-Johannessen, J.; Hansen, B.H.; Jefferis, B.; Fagerland, M.W.; Whincup, P.; Diaz, K.M.; Hooker, S.P.; Chernofsky, A.; et al. Dose-response associations between accelerometry measured physical activity and sedentary time and all cause mortality: systematic review and harmonised meta-analysis. BMJ 2019, 366.

5.        Tudor-Locke, C.; Craig, C.L.; Brown, W.J.; Clemes, S.A.; De Cocker, K.; Giles-Corti, B.; Hatano, Y.; Inoue, S.; Matsudo, S.M.; Mutrie, N.; et al. How many steps/day are enough? for adults. Int. J. Behav. Nutr. Phys. Act. 2011, 8, 1–17.
